# Unveiling the Potential of Diffusion Large Language Model in Controllable Generation

**Zhen Xiong**[1]    **Yujun Cai**[* 2,3]    **Zhecheng Li**[4]    **Yiwei Wang**[5]
[1]University of Southern California    [2]University of Queensland    [3]Ant Group
[4]University of California, San Diego    [5]University of California, Merced
eric2i.github.io/dLLM-CtrlGen

## Abstract

Controllable generation is a fundamental task in NLP with many applications, providing a basis for function calling to agentic communication. However, even state-of-the-art autoregressive Large Language Models (LLMs) today exhibit unreliability when required to generate structured output. Inspired by the current new diffusion-based large language models (dLLM), we realize that the architectural difference, especially the global information-sharing mechanism for language modeling, may be the key to unlock next-level controllable generation. To explore the possibility, we propose **S**elf-adaptive **S**chema **S**caffolding ($S^3$), a novel framework that enables dLLM to stably generate reliable structured outputs (e.g., JSON) by utilizing its innate reverse reasoning capability and global context awareness. $S^3$ initiates a schematic template directly in the output context as a starting state for dLLM, offering a more robust and general method than intricate prompt optimization. Experiments demonstrate that our method substantially unlocks the dLLM's potential in controllable generation in terms of structure adherence, content fidelity, and faithfulness. These results establish new perspectives and practical pathways for deploying language models in controllable generation tasks.

## 1 Introduction

Controllable generation is a fundamental task in the era of LLMs. It provides the foundation for stable tool use, agentic communication, and interaction with existing application programming interfaces (APIs). Existing works demonstrate that structured output still poses significant challenges even for state-of-the-art autoregressive LLMs. Many inspiring explorations have been conducted by previous researchers to address these challenges.

Prior autoregressive-based language model's methods pair a grammar-driven finite-state automata (FSA) with constrained decoding to enforce structural constraints during generation (Koo et al., 2024). When no token satisfies the grammar, all beams are pruned and generation halts. More broadly, as instruction following has improved, practitioners have turned to prompt-engineering heuristics to elicit structurally compliant outputs. Yet hand-crafted prompts for diverse structural specifications are labor-intensive and yield inconsistent results across domains and complexity levels.

Existing approaches share a fundamental limitation: they rely solely on language models' intrinsic capabilities without additional mechanisms to guide generation trajectories. This stems from architectural constraints of autoregressive models: (1) left-to-right generation prevents global structural coherence, as early tokens are generated without full sequence knowledge; (2) commitment to previously generated tokens limits backtracking when structural violations occur; (3) sequential dependencies inhibit simultaneous satisfaction of multiple constraints. Effective structured generation thus requires global sequence planning, iterative refinement, and parallel constraint satisfaction—capabilities that autoregressive architectures inherently lack.

---

[*]Corresponding author.

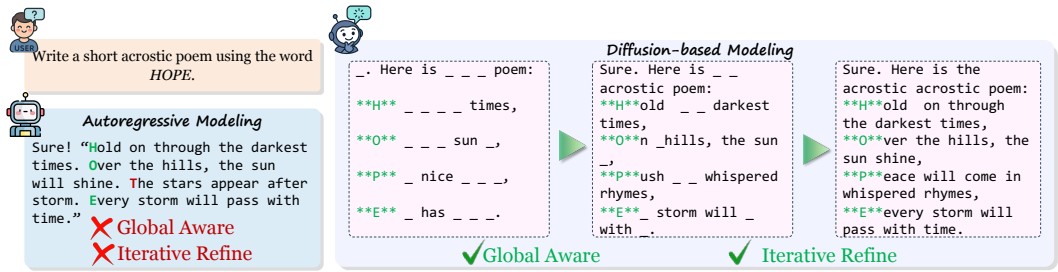

Figure 1: Illustrative comparison between autoregressive and diffusion-based language modeling on tasks requires specific global structure control and token-space planning in advance.

Under this scenario, we realize that the diffusion-based large language model (dLLM) can serve as a natural alternative to traditional autoregressive (AR) generation by iteratively denoising corrupted inputs, enabling global context modeling potential and parallel token generation (Arriola et al., 2025; Nie et al., 2025; Yu et al., 2025). Recent studies show that dLLMs can match AR models in instruction following, in-context learning, and math reasoning tasks (Gong et al., 2024; You et al., 2025), while also offering enhanced controllability and faster inference (Huang & Tang, 2025; Labs et al., 2025).

However, current open-sourced instruction-tuned dLLMs (Nie et al., 2025; Zhu et al., 2025; Ye et al., 2025b) fail to produce well-structured outputs, often generating hallucinated content or breaking structural constraints. Additionally, their inference speed remains limited. Analysis reveals that existing implementations (e.g., semi-autoregressive approaches Nie et al. (2025)) systematically undermine dLLMs' advantages in global awareness and parallel generation.

To unveil its full potential, we introduce a novel Self-adaptive Schema Scaffolding ($S^3$) method to fully unlock dLLM's potential in controllable generation. Specifically, $S^3$ inject a schematic template into the output context instead of instruction, providing a more *compelling* language prior for dLLM during generation. With our method, dLLM can achieve significantly improved structured output quality. To quantify performance, we introduce a comprehensive framework that evaluates structured outputs along three key dimensions: structural adherence, content fidelity, and faithfulness (with detailed definitions provided in Section 5). Experimental results show that our method marginally improves the performance structured output using dLLM compared with the commonly used prompting strategy.

Our main contributions can be summarized as follows: 1) We analyze the architectural advantages of diffusion-based large language models (dLLMs) compared with autoregressive models for controllable generation, focusing on prior's global attention mechanism and iterative refinement capabilities. 2) We propose Self-adaptive Schema Scaffolding ($S^3$), a training-free method that enables dLLMs to achieve higher structure output performance with fewer denoising steps. 3) We establish a comprehensive structure output evaluation framework focusing on structural adherence, content fidelity, and faithfulness metrics. Extensive experiments show that our $S^3$ achieves superior performance across all metrics with reduced computational complexity. Together, our work establishes a new perspective and practical solutions for deploying dLLMs for controllable generation tasks.

## 2 RELATED WORKS

**Structured Output** requires models to generate in predefined formats (e.g., code, JSON, XML, or tables), supporting tasks such as entity extraction (Li et al., 2024), classification Huang et al. (2025), and correlation prediction Xiong et al. (2025). Grammar-constrained decoding enforces compliance with context-free grammars (Geng et al., 2023) or type systems (Mündler et al., 2025) by adjusting next-token probabilities, without task-specific fine-tuning. Alternatively, planning-based or two-stage strategies first predict intermediate structures, such as abstract syntax trees, and then realize the final output (Wang et al., 2025). Most prior work relies on autoregressive LLMs for their strong language modeling and instruction-following ability (Wei et al., 2022). Diffusion-based models remain underexplored, motivating us to analyze their potential for structured generation and propose an evaluation framework covering structure compliance, content fidelity, and hallucination.

**Autoregressive Large Language Models**   Autoregressive LLMs have become a general solution across NLP tasks, strengthened by advances such as longer context windows (Liu et al., 2025), multimodal integration (Han et al., 2025), and test-time scaling (Jaech et al., 2024; Guo et al., 2025; Gemini Team, Google, 2025). They achieve state-of-the-art results on benchmarks including MMLU (Hendrycks et al., 2020), WebArena (Zhou et al., 2023), and AIME (Zhang, 2025), but still face persistent challenges in hallucination and controllability.

**Diffusion-based Language Models**   Diffusion models have recently been extended to discrete text, offering an alternative to autoregressive generation (Li et al., 2023). Research has introduced discrete score-based processes, refined noise schedules, and faster sampling methods, all aimed at improving efficiency and output quality. Building on these advances, large diffusion language models such as LLaDA (Nie et al., 2025; Zhu et al., 2025) and Dream (Ye et al., 2025b) demonstrate strong instruction-following ability. Yet their reliance on multi-step denoising weakens the advantage of parallel generation and slows inference (Israel et al., 2025). Motivated by these limitations, we focus on structured output as a setting where diffusion models can better exploit their architecture, and propose an inference pipeline that accelerates generation for practical applications.

## 3   PRELIMINARY AND NOTATIONS

### 3.1   AUTOREGRESSIVE LANGUAGE MODELING

Autoregressive large language models (LLMs) generate text by predicting the next token. During inference, instruction-tuned models (Ouyang et al., 2022; Wei et al., 2022) produce a response $A$ to a query $Q$ by sampling from the conditional distribution

$$\log P_\theta(A|Q) = \sum_{t=1}^{|A|} \log P_\theta(a_t|a_{<t}, Q),\tag{1}$$

where $A = (a_1, \ldots, a_{|A|})$, $a_t$ is the token at position $t$, $a_{<t}$ are the preceding tokens, $|A|$ is the sequence length, and $\theta$ are model parameters.

Due to the sequential nature of autoregressive generation, there are two systematic limitations: *models cannot directly access **future** token information during generation, and previously generated tokens cannot be **revised***. These constraints may limit the model's performance on tasks that would benefit from lookahead planning or iterative refinement.

### 3.2   DIFFUSION LANGUAGE MODELING

Diffusion models (Ho et al., 2020; Nichol & Dhariwal, 2021), originally developed for continuous image generation tasks, have been adapted to discrete language modeling (Austin et al., 2021a; Nie et al., 2025; Ye et al., 2025b).

Diffusion-based language models consist of two key processes: a forward noising process and a reverse denoising process. Given a tokenized sequence $\mathbf{x}_0$, the forward process progressively corrupts the sequence by masking tokens, producing increasingly noisy sequences $\mathbf{x}_t$ for $t \in [0, 1]$. As $t$ approaches 1, more tokens are masked until $x_1$ becomes a fully masked sequence. The reverse process trains a neural network to predict the original tokens at masked position within $x_t$ for $t \in (0, 1]$. The pre-training objective can be formulated as:

$$\min_\phi -\mathbb{E}_{t,x_0,x_t} \left[ \frac{1}{t} \sum_{i=1}^{L} \mathbf{1}[x_t^i = \mathbf{M}] \log P_\phi(x_0^i|x_t) \right]\tag{2}$$

where $\mathbf{M}$ denotes the mask token, $L$ is the sequence length, and $\phi$ represents the model parameters.

Recent research showed that diffusion large language model (dLLM) can be instruction-finetuned by concatenating ($\oplus$) user instruction $Q$ as fixed prefix to the masked target sequence $\mathbf{A}_t$ ($t = 1$, all masked initially), enabling conditioned generation. During inference, a single denoising step that the dLLM take can be factorized as:

$$\log P_\phi(Q \oplus A) = \sum_{i=1}^{|A|} \mathbf{1}[a_i = \mathbf{M}] \log P_\phi(a_i|Q \oplus A_t)\tag{3}$$

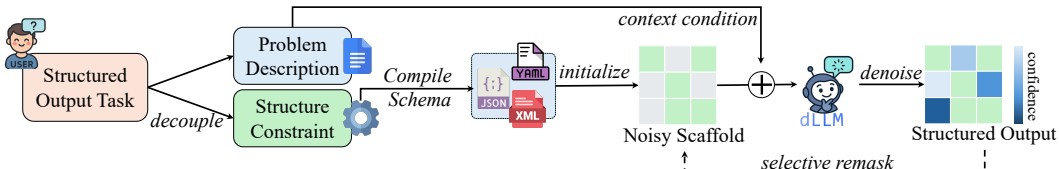

Figure 2: The overview of our method's pipeline. We begin by decomposing the original task instruction into two components: a problem description and a set of structural constraints. These constraints are compiled into a schema, which is then used to initialize a noisy scaffold where mask tokens serve as placeholders for missing content. The dLLM completes this scaffold by predicting the masked tokens, using the problem description as context to generate structured outputs. Additionally, we apply a selective remasking strategy that allows the model to iteratively refine its predictions and further improve generation quality.

Compared with AR models, the multi-step generation process can potentially be extended for iterative token edition and refinement (Havasi et al., 2025). More interestingly, the global attention mechanism of dLLM can largely improve its global context awareness and even including *future* planning capabilities (Ye et al., 2025b; 2024; 2025a) (See Appendix G). Therefore, in this paper, we exploit dLLM's superior global awareness and explore its potential for structured output.

# 4 METHODS

In this section, we first establish a rigorous theoretical framework for structured output generation (Section 4.1). Subsequently, we introduce our training-free pipeline, *Schema Scaffolding* ($S^2$) (Section 4.2) and its improved version *Self-adaptive Schema Scaffolding* ($S^3$)(Section 4.3).

## 4.1 TASK FORMALIZATION

Formally, we define structured output generation task as follows: given a user query $Q$, and a structural specification $S$, the objective is to generate a response $A$ that satisfies both the semantic requirements of $Q$ and the structural constraints defined by $S$. The structural specification $S$ can take various forms, including but not limited to: schema-based structures, format constraints, compositional structures, and domain-specific formats. The task can be mathematically formulated as a constrained optimization problem:

$$A^* = \arg \max_{A \in \mathcal{A}(S)} P_{\text{LM}}(A|Q, S) \tag{4}$$

where $\mathcal{A}(S)$ represents the space of all valid outputs conforming to structure $S$ and $P_{\text{LM}}$ is a conditional language model. In practice, searching entire $\mathcal{A}(S)$ is intractable due to the complexity of the token space.

## 4.2 SCHEMA SCAFFOLDING

Building on the theoretical insight we formulated in Section 3.2, we propose *schema scaffolding*, a training-free approach that explicitly incorporates structural constraints by pre-populating the generation context with structural templates.

The core idea of this approach is to transform unconstrained generation into a structured **fill-in-the-blank** task: In detail, our method operates by parsing the structural specification $S$ to identify invariant structural elements (e.g., brackets, delimiters, field names) and replacing variable content positions with mask tokens $\mathbf{M}$. This creates a structural scaffold $A_s$ that constrains the model's generation space while preserving semantic flexibility (See Appendix B for more details). To formalize why this approach is effective for diffusion models, we establish the following result:

**Theorem 4.1** (Scaffold-Guided Denoising Convergence)**.** *Let* $\mathbf{x}_0$ *be a target structured sequence and* $\mathbf{x}_t$ *be a partially masked sequence at timestep* $t$ *with scaffold* $\mathcal{S}$ *defining fixed structural positions. For a diffusion language model trained with objective (Eq. 2), initializing the denoising*

*process with structural scaffold $\mathcal{S}$ reduces the expected denoising error by:*

$$\mathbb{E}[\|\hat{\mathbf{x}}_0 - \mathbf{x}_0\|_{\mathcal{M}}] \leq \mathbb{E}[\|\tilde{\mathbf{x}}_0 - \mathbf{x}_0\|_{\mathcal{M}}] \cdot \left(1 - \frac{|\mathcal{S}|}{L}\right) \tag{5}$$

*where $\hat{\mathbf{x}}_0$ is generated with scaffolding, $\tilde{\mathbf{x}}_0$ is generated without scaffolding, $\|\cdot\|_{\mathcal{M}}$ denotes error over masked positions only, and $|\mathcal{S}|/L$ represents the scaffold coverage ratio.*

The proof is deferred to Appendix F.1. This result confirms that scaffolding provides a principled way to guide the denoising process, with error reduction proportional to the scaffold coverage. It also confirms our later empirical findings where even minimal scaffolding achieves near-perfect structural adherence.

Formally, the structured generation objective of our proposed method here is:

$$\begin{aligned} A^* &= \arg\max_{A \in \mathcal{A}(S)} P_{\text{LM}}(A|Q, S) \\ &\approx \arg\max_{A_s \in \mathcal{SC}} P_{\phi}(A_s|Q) \\ &= \arg\max_{A_s \in \mathcal{SC}} \sum_{a_i \in A_s} \mathbf{1}[a_i = \mathbf{M}] \log P_{\phi}(a_i|Q, A_s) \end{aligned} \tag{6}$$

Where $\mathcal{SC} \subset \mathcal{A}(S)$ represents the constrained subspace of outputs sharing the structural template derived from $S$, and $A_s$ denotes the scaffold with all non-masked tokens fixed except position $i$.

### 4.3 SELF-ADAPTIVE SCHEMA SCAFFOLDING

While our previous strategy constrains dLLM generation for improved structural compliance, it also introduces another challenge: *how to determine the appropriate number of mask tokens for each variable content position?* Although we can build structured fill-in-the-blank templates in advance, predicting the required length for each variable field remains problematic without prior knowledge of the target content.

A straightforward solution is to allocate ample mask tokens for each variable position, expecting the model to use only what is needed. Yet our analysis (Sec. 6.1) shows that dLLMs are sensitive to sequence length: longer scaffolds often distort generation quality instead of enabling selective usage, leading to under-utilization or hallucinated content. Another option is to introduce specialized padding tokens through fine-tuning. While appealing in principle, this violates our training-free objective and risks embedding dataset-specific biases that harm generalization to unseen domains.

Motivated by these intuitions, we propose an improved method that leverages the semantic token `null` as an placeholder. This approach preserves the training-free property while guiding dLLMs to naturally represent absent or variable-length content with `null` tokens.

Formally, we extend our scaffolding framework to incorporate adaptive length management:

$$A^* \approx \arg\max_{A_s \in \mathcal{SC}} \sum_{a_i \in A_s} \mathbf{1}[a_i = \mathbf{M}] \log P_{\phi}(a_i|Q^+, A_s) \tag{7}$$

where $Q^+$ represents our augmented prompt which guide the model to adopt `null` tokens to indicate absent values.

Our approach here empirically transforms the fixed-length scaffolding problem into an adaptive generation task where dLLMs can naturally handle variable-length fields and missing values. Further experimental results demonstrate that Self-Adaptive Schema Scaffolding significantly improves overall structured output quality, particularly in scenarios involving optional fields or variable-length content.

# 5 EXPERIMENTS

## 5.1 IMPLEMENTATION DETAILS

We use the LLaDA (Nie et al., 2025) model as our primary dLLM for experiments and the WikiBio dataset (Lebret et al., 2016) as our dataset. For reproducibility, we set the decoding temperature to zero during inference. See Appendix A for more details.

## 5.2 EVALUATION FRAMEWORK

Given the unique characteristics of structured output generation, we argue that traditional accuracy-based metrics alone are inadequate for capturing the full spectrum of model performance. Thus, we propose a comprehensive evaluation framework that assesses outputs across three key dimensions: **Structural Adherence**, **Content Fidelity**, and **Faithfulness**. This multi-dimensional benchmark enables a more nuanced and reliable assessment of a model's ability to generate coherent, informative, and trustworthy structured texts.

**Structural Adherence** measures how well-generated outputs conform to the target schema. Specifically, we define:

- *Structure Validity (SV)*: the proportion of outputs that are syntactically valid and parse without errors, capturing basic structural correctness.
- *Field Completeness (FC)*: the percentage of required fields that are correctly populated, indicating whether the model includes all mandatory schema components.
- *Schema Compliance (SC)*: the strictest structural metric, measuring the proportion of outputs that fully adhere to the predefined schema, including correct data types, value constraints, and nested structures.

**Content Fidelity** evaluates the semantic accuracy of information within structurally valid outputs. We consider:

- *Precision/Recall (PR/RE)*: precision and recall scores computed over individual field types, providing a detailed view of model behavior across content categories.
- *F1 Score (F1)*: the harmonic mean of field-level precision and recall, computed using both exact match and fuzzy match strategies to accommodate minor textual variations that preserve semantic meaning.

**Faithfulness** assesses the degree to which generated content remains grounded in the source input, which is especially critical in extraction settings.

- *Hallucination Rate (HR)*: the proportion of output fields that include information not present in or not reasonably inferable from the source text, directly reflecting the model's factual consistency.

Together, these metrics support systematic comparisons across models and highlight specific areas for targeted improvement in structured output generation.

## 5.3 MAIN RESULTS

This section presents a comprehensive evaluation of our approaches against baseline methods across three critical dimensions: structural adherence, content fidelity, and faithfulness.

### 5.3.1 STRUCTURAL ADHERENCE

Direct prompting dLLM is inadequate for structural constraint. Our baseline evaluation reveals performance consistently below 65% across all structural metrics—*Schema Validity*, *Field Completeness*, and *Schema Compliance*. Even with 32 denoising steps, the maximal score among these three critical metrics is above 87%, far lower than the expectation for realistic utility.(Fig. 3).

Our schema scaffolding methods marginally outperform the baseline method in terms of structural adherence. Both vanilla and self-adaptive variants enable near-perfect structural adherence with as few as 8 denoising steps, achieving performance saturation at 16 steps. This dramatic improvement

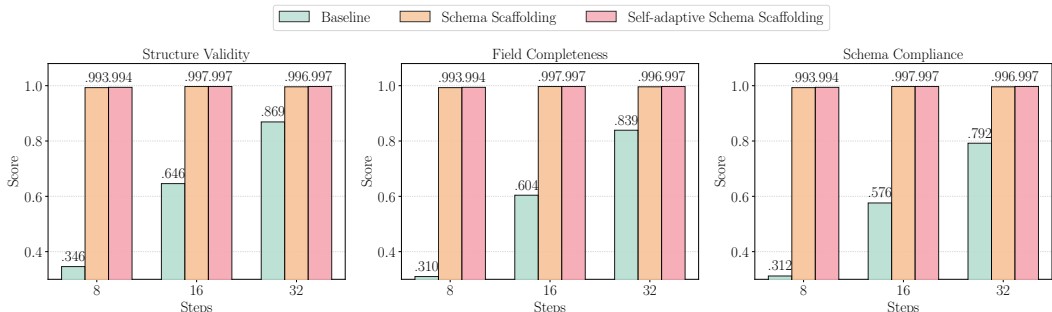

Figure 3: **Structural adherence** comparison across denoising steps and methods. Results show consistent improvements across all metrics using our schema scaffolding approaches, with near-perfect performance achieved in fewer steps.

carries practical significance beyond accuracy: since diffusion model inference scales linearly with denoising steps, our approach simultaneously reduces generation latency while enhancing structural quality. These findings establish that schema scaffolding addresses the fundamental barrier to deploying diffusion-based language models for structured output generation, elevating them from impractical to highly effective for real-world applications.

### 5.3.2 CONTENT FIDELITY

For content fidelity, additional denoising steps counterintuitively do not guarantee improved content accuracy and we observe that performance sometimes degrades with extended iteration (Fig. 4). This pattern reflects the diffusion model may deviation from optimal solutions during extended iterative reverse process, what we term the "overthinking" phenomenon.

Vanilla Schema Scaffolding demonstrates clear improvements in recall and F1 score relative to baseline, indicating enhanced coverage of relevant content. However, precision suffers notably across all denoising configurations. This trade-off emerges from the model's compensatory behavior: when constrained by rigid schema requirements, it over-generates tokens to fill all reserved slots, particularly problematic when content length varies significantly across examples.

Our Self-adaptive Schema Scaffolding ($S^3$) resolves this tricky situation. By incorporating adaptive `null` token for surplus slots, the method prevents over-generation while maintaining comprehensive coverage. This simple yet effective recipe yields substantial improvements across all three metrics, establishing robust content fidelity without sacrificing structural compliance.

| Method | Computation Budget | | |
|---|---|---|---|
| | 8 Steps | 16 Steps | 32 Steps |
| Baseline | 0.404 | 0.403 | 0.409 |
| $S^2$ | 0.465 | 0.463 | 0.463 |
| $S^3$ | **0.340** | **0.331** | **0.331** |

Table 1: Hallucination rate (*lower is better*) comparison across various denoising steps. Our self-adaptive schema scaffolding ($S^3$) method consistently achieves the lowest hallucination rates, indicating superior faithfulness. Boldface indicates the best results.

### 5.3.3 FAITHFULNESS

Our self-adaptive schema scaffolding ($S^3$) method demonstrates superior faithfulness performance, consistently achieving the lowest hallucination rates across all denoising steps (Tab. 1). This clear advantage over baseline approaches establishes the superiority of our method in maintaining factual grounding while generating structured outputs.

Interestingly, vanilla schema scaffolding reveals elevated hallucination rates compared to baseline methods. The worse performance in faithfulness represents a critical weakness that challenges our original design of the Schema Scaffolding approach. Our analysis reveals that this hallucination issue stems from the distributional shifts introduced by the imposed structural constraints, which create a token distribution that diverges from the model's pre-training reverse process. Specifi-

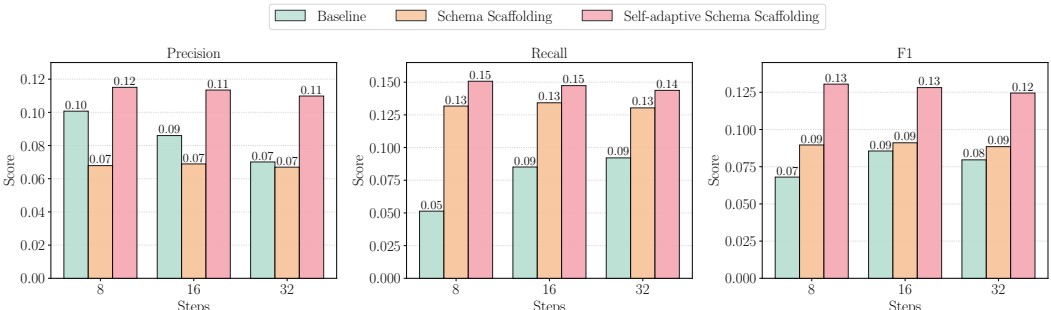

Figure 4: **Content fidelity** comparison across denoising steps and methods. Our self-adaptive schema scaffolding consistently achieves the highest precision, recall, and F1 score across all settings.

cally, the rigid schema acts as a structural prior that misaligned with the diffusion language model's learned denoising process. This distributional mismatch forces the model into suboptimal denoising trajectories. When the structural prior demands content generation beyond what can be grounded in the source text, the model defaults to fabricating plausible tokens to satisfy schema requirements—a behavior that contradicts its training objective of faithful reconstruction.

Our self-adaptive approach mitigates this fundamental conflict by allowing the model to acknowledge missing information through `null` tokens, thus maintaining alignment with its pre-trained denoising capabilities while respecting structural constraints.

### 5.3.4 ABLATION STUDY

To demonstrate the effectiveness of our approach, we conduct ablation studies comparing our method against baseline models that are incrementally enhanced with different techniques (Tab.2). While few-shot learning and template-as-guidance approaches improve dLLM's structural adherence and faithfulness, their performance on fidelity remains inconsistent and shows limited improvement. In contrast, our zero-shot method achieves superior data efficiency and computational efficiency. We also observe strong diminishing returns beyond 8 denoising steps, indicating that the structural constraints provided by the scaffold are realized rapidly during inference. Despite requiring fewer denoising steps, $S^3$ delivers remarkably stable structural adherence, along with marginal improvements in both fidelity and faithfulness. We also tested additional datasets to validate the generalizability of our approach (See Appendix D more results).

## 6 DISCUSSION

### 6.1 ROBUSTNESS AGAINST HALLUCINATION

Similar to autoregressive language models, diffusion-based language models (dLLMs) also suffer from hallucination—generating nonfactual content, faulty reasoning, or unsupported conclusions. Unlike open-ended dialogue or creative text generation, hallucination in structured output is particularly detrimental, as it directly undermines the reliability and trustworthiness of the output—qualities that are central to these tasks.

Our vanilla schema scaffolding method enforces a strong structural constraint, which inevitably interferes with the natural generation trajectory of dLLMs. To mitigate the resulting hallucination, we observe that it is possible to transform unfamiliar test-time scenarios into familiar training-time cases by providing sufficient in-context guidance. Thus, we introduce the notion of a special token, `null`, as a flexible placeholder. Once the model adapts this convention, it can fill otherwise empty slots without resorting to fabricated content, despite not being explicitly trained to use padding tokens. This simple yet effective prior guidance enables dLLMs to substantially reduce hallucination, and it inspired the formulation of our improved $S^3$ method.

| Experiment | Structural Adherence | | | Content Fidelity | | | Faithfulness |
|---|---|---|---|---|---|---|---|
| | SV↑ | FC↑ | SC↑ | PR↑ | RE↑ | F1↑ | HR↓ |
| 8 Steps | | | | | | | |
| baseline | 0.346 | 0.310 | 0.312 | 0.101 | 0.051 | 0.068 | 0.404 |
| w/ few-shots | 0.471 | 0.447 | 0.443 | 0.086 | 0.056 | 0.068 | 0.366 |
| w/ template | 0.475 | 0.432 | 0.431 | 0.088 | 0.081 | 0.084 | 0.388 |
| $S^3$ (ours) | **0.994** | **0.994** | **0.994** | **0.115** | **0.151** | **0.130** | **0.340** |
| 16 Steps | | | | | | | |
| baseline | 0.646 | 0.604 | 0.576 | 0.086 | 0.085 | 0.086 | 0.403 |
| w/ few-shots | 0.735 | 0.713 | 0.674 | 0.084 | 0.087 | 0.085 | 0.371 |
| w/ template | 0.794 | 0.734 | 0.738 | 0.091 | 0.139 | 0.110 | 0.390 |
| $S^3$ (ours) | **0.997** | **0.997** | **0.997** | **0.113** | **0.147** | **0.128** | **0.331** |
| 32 Steps | | | | | | | |
| baseline | 0.869 | 0.839 | 0.792 | 0.070 | 0.092 | 0.080 | 0.409 |
| w/ few-shots | 0.890 | 0.870 | 0.824 | 0.092 | 0.114 | 0.101 | 0.358 |
| w/ template | 0.909 | 0.886 | 0.882 | 0.091 | **0.160** | 0.116 | 0.384 |
| $S^3$ (ours) | **0.997** | **0.997** | **0.997** | **0.110** | 0.144 | **0.125** | **0.331** |

Table 2: Ablation study results comparing different experimental configurations. Boldface indicates the best results. We use 3 examples for few-shot learning (Wei et al. (2022)). To follow previous practice, we also extend the instruction with a complete schema (Wang et al. (2025)) as guidance. Our method consistently outperforms these alternative techniques, indicating a nontrivial and superior improvement.

## 6.2 COMPLEXITY ANALYSIS

An existing bottleneck of dLLMs lies in their inference speed. Empirically, the multi-step denoising process introduces a latency that grows positively with the number of diffusion steps. Within each step, the global attention computation incurs a quadratic cost with respect to the total context length $L$. As a result, the overall computational complexity of the reverse process scales as $\mathcal{O}(L^3)$. To mitigate this cost, some implementations adopt a semi-autoregressive decoding scheme with block-wise KV-caching, which partially reduces the computational burden. However, this design still compromises the core parallelism advantages of diffusion-based decoding.

For structured generation tasks, where the output structure is known or can be approximated, our proposed method $S^3$ introduces an alternative perspective and initialize the reverse process from a partially denoised state rather than a fully random one. This *warm-start* initialization serves as a structural prior, effectively providing a language scaffold that accelerates generation and enhances controllability. Asymptotically, $S^3$ reduces the decoding complexity to $\mathcal{O}(nL^2)$, where $n$ is a tunable hyperparameter that remains significantly smaller than $L$ in practice.

## 7 CONCLUSION

In this paper, we explore the potential of diffusion large language models' global awareness for controllable generation of structured output. We propose the novel Self-adaptive Schema Scaffolding method ($S^3$) that guides dLLMs to adaptively generate fully controllable structured output by manipulating the reverse process and leveraging the innate global attention mechanism. Our comprehensive evaluation framework demonstrates that $S^3$ achieves superior structural adherence, content fidelity, and reduced hallucination rates. Through complexity analysis, we show that our approach maintains computational efficiency while enabling higher levels of controllability and hallucination control. We believe these findings establish dLLMs as a promising alternative for controllable generation tasks.

ACKNOWLEDGMENTS

The work is partially supported by the U.S. National Science Foundation (NSF) Grant CRII 2451683, an NVIDIA Academic Grants Program, a U.S. Bank Academic Research Award, the University of California, Merced, and a UC Merced Faculty Research Award. The views and conclusions are those of the authors and do not necessarily reflect the official policy or position of the U.S. Government.

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

## A  IMPLEMENTATION DETAILS

We use the `GSAI-ML/LLaDA-1.5` diffusion large language model, loaded from HuggingFace official with `bfloat16` for optimal memory efficiency and performance. All experiments were conducted on one single NVIDIA RTX 4090 GPU. We set `max_new_tokens=128`, which suffices all structured output task, and `temperature=0` for reproducibility. The WikiBio dataset serves as the primary benchmark dataset.

## B  SCHEMA COMPILING

The compilation step serves as the bridge between the user's structural constraints and the diffusion model's initial state. Formally, given a schema $\mathcal{S}$ and a target format $\mathcal{F}$ (e.g., JSON, XML, YAML), the compiler function $C(\mathcal{S}, \mathcal{F})$ generates a scaffold sequence $\mathbf{x}_{\text{scaffold}}$. A key advantage of $S^3$ is that the diffusion process is agnostic to the underlying data format. As long as a standard parser exists for a given format, the schema can be compiled into a scaffold. We demonstrate this universality below with JSON and XML examples.

**Case 1: JSON Compilation**  For a Biography generation task, the JSON schema defines fields for name and birth year.

**Input Schema (JSON):**

```
{
  "type": "object",
  "properties": {
    "name": {"type": "string"},
    "born": {"type": "integer"}
  }
}
```

**Compiled Scaffold ($S^3$ Initial State):**

```
{
  "name": [MASK]...[MASK],
  "born": [MASK]...[MASK]
}
```

**Case 2: XML Compilation**  Applying the same logic to a hierarchical format like XML requires no changes to the diffusion algorithm, only to the initial compiling parser.

**Input Schema (XML XSD style):**

```
<xs:element name="person">
  <xs:element name="name"
              type="xs:string"/>
  <xs:element name="born"
              type="xs:int"/>
</xs:element>
```

**Compiled Scaffold ($S^3$ Initial State):**

```
<person>
  <name>[MASK]...[MASK]</name>
  <born>[MASK]...[MASK]</born>
</person>
```

## C  REMASKING STRATEGY

By default, dLLMs can generate all tokens within the full context in parallel through a single inference step. However, for many general NLP tasks, this strategy is often suboptimal. For example, simultaneously generating all tokens that form a complete solution to a math problem is significantly harder than doing so incrementally via the autoregressive approach. To address this issue, prior works have explored various remasking strategies to bridge the gap.

One common strategy is the block-wise masking, which employs a sliding-window mechanism. The context is divided into blocks, and at each iteration, only the tokens within the current window are taken into consideration, while those in following blocks are remasked for future regeneration. Thus, this strategy is also considered semi-autoregressive. However, empirical results show that using a block size of one, which essentially reverting to a fully autoregressive process, often yields the best performance. We consider such a strategy largely eliminates the parallelism advantage of diffusion-based models.

Another type of works focus on low-confidence remasking, which selectively discards tokens with low confidence scores, such as low log-probability or high entropy, relative to others in the current

iteration. This process is repeated until all tokens are finalized, allowing for a more adaptive number of iterations. Some approaches further combine low-confidence filtering with block-wise remasking to support in-block autoregressive generation.

In contrast, our proposed $S^3$ method adopts a simple yet effective top-$K$ remask strategy, where $K = O/n$, with $O$ denoting the total number of tokens to be generated and $n$ being a tunable number of denoising steps. Compared to the block-wise remasking method, our approach is significantly more efficient and introduces an additional level of controllability by predefining the number of generation iterations, offering a practical trade-off between speed and output quality.

| Remasking Strategy | Steps | SV↑ | FC↑ | SC↑ | F1↑ | HR↓ |
|---|---|---|---|---|---|---|
| **Top-K** ($S^3$) | 16 | **0.997** | **0.997** | **0.997** | **0.128** | **0.331** |
| Low-Confidence | 16 | 0.994 | 0.993 | 0.992 | 0.121 | 0.348 |
| Block-wise | 16 | 0.989 | 0.986 | 0.985 | 0.115 | 0.362 |

Table 3: Ablation of different remasking strategies on structured generation (WikiBio, 16 steps).

Additionally, our additional experiment in Table 3 indicate that **Top-K** approach, adopted by our method, yields superior performance across all structural (SV, FC, SC) and content (F1, HR) metrics. We attribute this to the deterministic nature of Top-K remasking, which provides precise iteration control ($K_t = O/n$). Unlike Low-Confidence remasking, which may re-mask too few or too many tokens in early steps leading to unstable convergence, Top-K ensures a predictable computational budget and steady refinement of the scaffolded structure. Block-wise remasking performed the worst, suggesting that structural constraints in formats like JSON often require global, non-contiguous corrections rather than local block updates.

# D ADDITIONAL GENERALIZABILITY EXPERIMENTS

To validate the effectiveness of our approach beyond the primary WikiBio dataset, we conducted extensive evaluations across diverse domains and model architectures. This section details the performance of $S^3$ on code generation, dialogue state tracking, and form filling, as well as its transferability to other diffusion language models.

## D.1 CROSS-DOMAIN EVALUATION

We extended our evaluation to three additional datasets that require distinct structural formats beyond standard JSON:

- **Code Generation (MBPP)** (Austin et al., 2021b): Using the Mostly Basic Python Problems dataset, requiring strict syntactic adherence to Python code structures.
- **Dialogue Structure (MultiWOZ)** (Budzianowski et al., 2018): Evaluating dialogue state tracking, which requires structured output representing user intent and slot values.
- **Form Filling (FUNSD)** (Jaume et al., 2019): A visual document understanding task requiring the extraction of key-value pairs in structured formats.

As shown in Table 4, $S^3$ demonstrates consistent improvements across all tasks. We observed absolute gains of $15 \sim 35\%$ in structural adherence metrics (SV, FC, SC) compared to the baseline LLaDA model. This confirms that the scaffolding mechanism generalizes effectively to different structural constraints (e.g., Python indentation, dialogue states) without task-specific tuning.

## D.2 CROSS-ARCHITECTURE EVALUATION

To ensure our findings are not specific to a single architecture, we validated $S^3$ on Dream-7B (Ye et al., 2025b), another state-of-the-art diffusion language model. Using the same hyperparameters described in Section 5, we compared the baseline generation against our scaffolding method.

The results in Table 5 show that $S^3$ boosts the Structural Validity (SV) of Dream-7B from 0.723 to 0.991. This indicates that our method exploits the fundamental non-autoregressive nature of diffusion models rather than model-specific artifacts.

| Task | Dataset | Method | Structural Metrics (SV / FC / SC) |
|------|---------|--------|-----------------------------------|
| Code Generation | MBPP | LLaDA Baseline
**LLaDA + S$^3$** | 0.645 / 0.618 / 0.592
**0.989 / 0.991 / 0.988** |
| Dialogue Structure | MultiWOZ | LLaDA Baseline
**LLaDA + S$^3$** | 0.712 / 0.684 / 0.651
**0.982 / 0.985 / 0.981** |
| Form Filling | FUNSD | LLaDA Baseline
**LLaDA + S$^3$** | 0.701 / 0.673 / 0.649
**0.994 / 0.993 / 0.992** |

Table 4: Performance comparison on diverse structured output tasks (Code, Dialogue, and Forms) using the LLaDA model. $S^3$ significantly outperforms the baseline across Structural Validity (SV), Format Compliance (FC), and Schema Consistency (SC).

| Method | SV | FC | SC | F1 |
|--------|-----|-----|-----|-----|
| Dream-7B (Baseline) | 0.723 | 0.698 | 0.671 | 0.082 |
| **Dream-7B + S$^3$** | **0.991** | **0.989** | **0.988** | **0.121** |

Table 5: Generalization of $S^3$ to the Dream-7B diffusion model.

## E  PROMPTS

This section provides the complete prompts used in our experiments for both the baseline method and our proposed Self-adaptive Schema Scaffolding ($S^3$) approach.

The baseline method employs a detailed prompt that explicitly specifies the desired output structure through a comprehensive JSON schema, as shown in Figure 5. This approach requires the model to understand and adhere to the predefined schema constraints based solely on the textual instructions.

In contrast, our Self-adaptive Schema Scaffolding ($S^3$) method utilizes a significantly simplified prompt (Figure 6) that omits explicit structural specifications. Instead, our method relies on the inherent structural constraints enforced by the scaffolding mechanism, demonstrating the token efficiency and clarity of $S^3$.

## F  PROOF

### F.1  PROOF OF THEOREM. 4.1

*Proof.* The diffusion model learns to predict $p_\phi(x_0^i|x_t)$ for each masked position $i$ where $x_t^i = \mathbf{M}$. With structural scaffolding, we partition positions into: - Fixed scaffold positions: $\mathcal{S} = \{i : x_t^i \neq \mathbf{M}, \text{ fixed by structure}\}$ - Variable positions: $\mathcal{V} = \{i : x_t^i = \mathbf{M}\}$

The model's prediction at each masked position depends on the context:

$$p_\phi(x_0^i|x_t) = p_\phi(x_0^i|\{x_t^j\}_{j \neq i})$$

With scaffolding, the context includes correct structural tokens at positions $\mathcal{S}$, providing stronger signal:

$$p_\phi(x_0^i|x_t \text{ with } \mathcal{S}) = p_\phi(x_0^i|\mathcal{S} \cup \{x_t^j\}_{j \in \mathcal{V} \setminus \{i\}})$$

Since the scaffold tokens are correct by construction (they match the target structure), they reduce uncertainty in the conditional distribution. The error reduction is proportional to the informativeness of the scaffold.

For each masked position, the expected error with scaffolding is:

$$\mathbb{E}[\epsilon_i|\mathcal{S}] \leq \mathbb{E}[\epsilon_i] \cdot (1 - I(\mathcal{S}; x_0^i))$$

---

**Baseline Prompt**

**Instruction:**
Extract the following information from the provided document and return only a JSON response with no additional text or explanation: name, birth_date, birth_place, death_date, death_place, nationality, occupation.
The response must conform to this JSON schema:

```
{
  "schema": "http://json-schema.org/draft-07/schema\#",
  "type": "object",
  "properties": {
    "name": {
      "type": ["string", "null"],
      "description": "Full name of the person"
    },
    "birth\_date": {
      "type": ["string", "null"],
      "description": "Birth date in ISO format (YYYY-MM-DD, YYYY-MM, or YYYY)"
    },
    "birth\_place": {
      "type": ["string", "null"],
      "description": "Place of birth (city, country format preferred)"
    },
    "death\_date": {
      "type": ["string", "null"],
      "description": "Death date in ISO format (YYYY-MM-DD, YYYY-MM, or YYYY)"
    },
    "death\_place": {
      "type": ["string", "null"],
      "description": "Place of death (city, country format preferred)"
    },
    "nationality": {
      "type": ["string", "null"],
      "description": "Nationality or citizenship"
    },
    "occupation": {
      "type": ["string", "null"],
      "description": "Primary occupation or profession"
    }
  },
  "required": ["name", "birth\_date", "birth\_place", "death\_date", "death\_place", "nationality", "
      occupation"]
}
```

If any information is not available in the document, use null for that field.
[*DOCUMENT TEXT*]

Figure 5: The complete prompt we use for the baseline method.

---

$S^3$ **Prompt**

**Instruction:**
Extract information from the provided document and return only a JSON response with no additional text or explanation. If any information is not available in the document, use `null` for that field. [*DOCUMENT TEXT*]

---

Figure 6: The complete prompt for our Self-adaptive Schema Scaffolding ($S^3$) method. Unlike the baseline approach, our method does not require explicit structural information in the prompt, as it naturally enforces predefined structure constraints on the model's output.

where $I(\mathcal{S}; x_0^i)$ is the mutual information between scaffold and target token.

Aggregating over all masked positions and noting that structural tokens typically have high mutual information with content tokens (e.g., field names predict value types), we get:

$$\mathbb{E}[\|\hat{\mathbf{x}}_0 - \mathbf{x}_0\|_{\mathcal{M}}] \leq \mathbb{E}[\|\tilde{\mathbf{x}}_0 - \mathbf{x}_0\|_{\mathcal{M}}] \cdot \left(1 - \frac{|\mathcal{S}|}{L}\right)$$

The equality holds when scaffold and content are independent, which rarely occurs in structured generation. $\qquad \square$

## G    PLANNING ABILITY OF DLLMS

The future planning capability is an architectural advantage of diffusion large language models due to the global awareness as we formalized in Section 3. Previous works (Ye et al., 2024; 2025a) have shown that text diffusion models exhibit superior planning capabilities in small-scale, task-specific contexts. Additionally, another work (Ye et al., 2025b) has also empirically proven that dLLM is significantly better than similar-scale AR models like (Llama3 and Qwen2.5) in benchmarks requiring planning ahead, such as *trip planning*, *constrained arithmetic problems*, and *Sudoku*.

## H    THE USE OF LARGE LANGUAGE MODELS (LLMS)

We used an LLM to assist with the phrasing and grammar of the manuscript. The LLM was used strictly as a writing aid and did not contribute to the scientific ideation, methodology, or results presented in this paper.

