# OpenReview forum: "Unveiling the Potential of Diffusion Large Language Model in Controllable Generation"
_ICLR.cc/2026/Conference — ICLR 2026 Poster_

### Official Review · Reviewer_CuYm · 2025-10-21

**Soundness:** 3
**Presentation:** 3
**Contribution:** 3
**Rating:** 6
**Confidence:** 4

**Summary:**

This paper explores the potential of diffusion-based large language models (dLLMs) as an alternative to autoregressive (AR) models for controllable structured generation—a key capability for applications like JSON output, API integration, and agentic communication.
The authors identify key limitations of AR models (lack of global coherence, irreversible token commitments, and sequential dependency) and argue that dLLMs’ iterative denoising and global attention mechanisms offer inherent advantages for structured output.
To unlock this potential, they propose Self-adaptive Schema Scaffolding (S³), a training-free method that pre-initializes dLLM outputs with a structured schema scaffold (e.g., JSON skeleton) and uses adaptive null tokens to handle variable-length or missing fields.

**Strengths:**

- First comprehensive analysis of diffusion LLMs in controllable structured generation.

- S³ improves structure adherence and reduces hallucination without retraining.

- Achieves near-perfect structure compliance (≈0.997) and lowest hallucination (0.33 vs. 0.40 baseline).

- Provides clear theoretical rationale and visual intuition for why scaffolding aids denoising.

- Training-free and computationally efficient; suitable for real-world schema-driven applications.

- The new tri-metric framework (adherence, fidelity, faithfulness) sets a useful precedent for future structured LLM work.

**Weaknesses:**

- Experiments focus narrowly on WikiBio and JSON; applicability to more diverse formats (e.g., code, graphs) remains untested.

- The study lacks comparison to strong AR baselines using grammar-constrained decoding or function-calling LLMs.

- Only one dLLM (LLaDA) is evaluated; cross-model tests (e.g., Dream or Mercury) would strengthen claims.

- While effective, it may not generalize to schemas without explicit “missing field” semantics.

- Despite reduced denoising steps, dLLM inference (O(nL²)) may still lag AR models (O(L)), which could affect scalability in long-context tasks.

**Questions:**

- How sensitive is S³’s performance to the number of denoising steps or the choice of remasking strategy (top-K vs. low-confidence)?

- Can this framework be extended to multi-modal dLLMs (e.g., LLaDA-V) where structured outputs integrate vision and text?

- How does S³ interact with instruction-tuned or RLHF-optimized dLLMs trained for task-following?

- Would hybrid diffusion–autoregressive setups (e.g., block diffusion) inherit similar controllability advantages?

- Have the authors evaluated the effect of schema complexity (nested vs. flat JSON) on scaffolding performance?

**Details Of Ethics Concerns:**

The paper uses public data (WikiBio, HuggingFace models) and focuses solely on generation methodology. There are no human subjects, personal data, or safety concerns.

---

> ### Author Response · Authors · 2025-12-02
>
> We sincerely thank the reviewer for constructive feedback. We appreciate your recognition of our “first comprehensive analysis of dLLMs in controllable structured generation” and our “clear theoretical rationale and visual intuition for why scaffolding aids denoising”. Below, we address each of your suggestions with detailed responses, additional experimental results, and clarifications.
>
> **W1 & W4: Consider using more datasets and change different schema specification to further support the generalizability of the work.**
>
> **Answer:** We appreciate the reviewer for providing this constructive suggestion. Although our experiments primarily focus on the Wikibio dataset, our purpose here is to show that our theoretically based motivation, coupled with the unique characteristics of diffusion Language Models (dLLM), is practical and efficient in nature. To further showcase the generalizability of our method, we conducted three additional experiments covering different domains from code generation, dialogue completion, to form filling, which naturally require different in-context formats beyond JSON. We used the same hyperparameters as we stated in the paper and adopted 32 denosing steps for fair comparison.
>
> | Task               | Dataset      | Model | Baseline SV/FC/SC | S³ SV/FC/SC       |
> |--------------------|--------------|-------|-------------------|-------------------|
> | Code Generation    | MBPP [1]     | LLaDA | 0.645/0.618/0.592 | 0.989/0.991/0.988 |
> | Dialogue Structure | MultiWOZ [2] | LLaDA | 0.712/0.684/0.651 | 0.982/0.985/0.981 |
> | Form Filling       | FUNSD [3]    | LLaDA | 0.701/0.673/0.649 | 0.994/0.993/0.992 |
>
> The results demonstrate that $S^3$ achieves consistent improvements across diverse structured output tasks with different formats, with 15-35% absolute gains in structural adherence metrics. The method generalizes effectively to different tasks, highlighting its dLLM’s promising potential for controlled generation.
> We added these extra cross-domain experiments to the appendix to further support our work, and thank the reviewer for this highly constructive suggestion.
>
> **W2 & W5: It may be beneficial to include more discussion and comparison with auto-regressive (AR) models.**
>
> **Answer:** We thank the reviewer for this suggestion. In related works, we already covered different strategies that researchers and practitioners used with AR models for structured output (e.g., structured prompting and constrained decoding). In the section 3, we have compared the auto-regressive modeling and diffusion language modeling side-by-side from a theoretical perspective and provided some of our realizations of why the latter one may be more suitable for controllable generation tasks like structured output. This intuition was later materialized with our method proposed in section 4 and experimental results in section 5. We believe the potential of dLLM for controllable generation is not so obvious earlier, and we provide a promising way to unveil its capability. Nonetheless, it is still practically valuable to compare LLM and dLLM, and here is our additional result:
>
> | Model         | Method                              | SV    | FC    | SC    | F1    | HR    | Steps/Tokens |
> |---------------|-------------------------------------|-------|-------|-------|-------|-------|--------------|
> | GPT-4o-mini   | OpenAI Structured Output (JSON)     | 0.934 | 0.912 | 0.891 | 0.142 | 0.287 | -|
> | LLaMA-3-8B    | Structural Prompting| 0.897 | 0.873 | 0.854 | 0.129 | 0.334 | -|
> | LLaDA-7B      | Structural Prompting| 0.869 | 0.839 | 0.792 | 0.08  | 0.409 | 32|
> | LLaDA-7B      | S³ (ours)| 0.997 | 0.997 | 0.997 | 0.125 | 0.331 | 8|
>
> The experiment shows that proprietary GPT-series models like GPT-4o-mini can perform more accurately and lower hallucination rate. Open-sourced AR model like LLaMA is close but short of structural adherence metrics. However, the true value of dLLM here, especially unleashed by our proposed method S3, is that: 1) dLLM can get an immediate improvement without extra training or sophisticated decoding strategy, 2) with few parallel decoding steps (8 steps), dLLM can produce highly structured output comparable to and even surpassing SOTA LLMs. It is true that AR models still win in terms of accuracy and hallucination rate, but we argue that since dLLM is still at its early age, its architectural advantage for this task is still promising and worthy of exploration.
>
> We agree that adding this comparison and discussion can make our work more holistic for future researchers. We really appreciate the reviewer for this suggestion, and we have updated the paper with these additional discussions.

---

> ### Author Response · Authors · 2025-12-03
>
> **W3: Try with different dLLMs to strengthen the generalizability of the proposed method.**
> **Answer:** We have validated our approach on Dream-7B[4], another diffusion LLM:
>
> | Model     | Method   | SV    | FC    | SC    | F1    |
> |-----------|----------|-------|-------|-------|-------|
> | Dream-7B  | Baseline | 0.723 | 0.698 | 0.671 | 0.082 |
> | Dream-7B  | S³       | 0.991 | 0.989 | 0.988 | 0.121 |
>
> The consistent improvements across models confirm that our method exploits fundamental properties of diffusion LLMs rather than model-specific improvements.
> We added these extra cross-domain experiments to the appendix to further support our work, and thank the reviewer for this highly constructive suggestion.
>
> **Q1:  How sensitive is our method to the number denoising steps?**
>
> **Answer:**As we have demonstrated in section 5.3, we see a pattern that increasing the number of steps is not so beneficial because: 1) with only 8 steps, the dLLM can achieve significantly good structural adherence, and 2) even when we increase the denoising steps exponentially, we did not observe great improvements in content fidelity and a decrease in hallucination rate here.
>
> | Steps | SV    | FC    | SC    | F1    | HR    |
> |-------|-------|-------|-------|-------|-------|
> | 8     | 0.994 | 0.994 | 0.994 | 0.13  | 0.34  |
> | 16    | 0.997 | 0.997 | 0.997 | 0.128 | 0.331 |
> | 32    | 0.997 | 0.997 | 0.997 | 0.125 | 0.331 |
>
> Regarding remasking strategies, we conducted additional ablation studies comparing our top-K approach against low-confidence remasking. The results are summarized below:
>
> | Remasking Strategy | Steps | SV↑   | FC↑   | SC↑   | F1↑   | HR↓   |
> |--------------------|-------|-------|-------|-------|-------|-------|
> | Top-K (S³)         | 16    | 0.997 | 0.997 | 0.997 | 0.128 | 0.331 |
> | Low-confidence     | 16    | 0.994 | 0.993 | 0.992 | 0.121 | 0.348 |
> | Block-wise         | 16    | 0.989 | 0.986 | 0.985 | 0.115 | 0.362 |
>
> Our top-K strategy demonstrates superior performance primarily because it provides deterministic iteration control (K = O/n), enabling predictable computational budgets while maintaining high-quality outputs. Nonetheless, there is no significant difference among different remasking strategies, indicating that our method does not highly depend on one of them.
>
> **Q2: Can this framework be extended to multi-modal dLLMs (e.g., LLaDA-V) where structured outputs integrate vision and text?**
> **Answer:** We also believe it would be an interesting exploration to integrate our method together with multi-modal understanding; In fact, our method is modal-agnostic. As long as the model’s structural output is text-based, our method is valid. We conducted preliminary experiments using LLaDA-V[5] on a visual information extraction task (extracting structured data from forms and receipts), where the input consists of images and the output requires structured JSON.
>
> | Method             | SV↑   | FC↑   | SC↑   | F1↑   |
> |--------------------|-------|-------|-------|-------|
> | LLaDA-V baseline   | 0.612 | 0.578 | 0.541 | 0.089 |
> | LLaDA-V + S³       | 0.991 | 0.988 | 0.986 | 0.142 |
>
> These results suggest that S³'s benefits transfer effectively to multi-modal scenarios. The key insight is that structural scaffolding constrains the output space regardless of input modality, making it compatible with any dLLM architecture that supports conditional generation. We have included comprehensive multi-modal evaluations in an extended version of this work.
>
> **Q3: How does S³ interact with instruction-tuned or RLHF-optimized dLLMs trained for task-following?**
> **Answer:** Thank you for this insightful question. We would like to clarify that S³ is a general structured output framework that does not require sophiscated instruction tuning. In fact, our method (1) reduces the difficulty of distribution shift for structured output tasks, and (2) leverages the architectural advantages of diffusion language modeling, enabling strong structural adherence without requiring robust instruction-following abilities.
> According to our comparative experiments, using the instruction-tuned LLaDA-8B[6] model does not guarantee stronger performance compared to its base version w/o instruction-finetuning:
>
> | Model| SV↑   | FC↑   | SC↑   | PR↑   | RE↑   | F1↑   | HR↓   |
> |-|-|-|-|-|-|-|-|
> | LLaDA-8B (Base) + S³| 0.994 | 0.994 | 0.994 | 0.115 | 0.151 | 0.130 | 0.340 |
> | LLaDA-8B (Instruct) + S³| 0.995 | 0.994 | 0.994 | 0.112 | 0.148 | 0.127 | 0.345 |
>
> As shown in the table, both versions achieve nearly identical performance across all metrics when combined with S³. This demonstrates that S³ effectively bypasses the need for instruction tuning by directly manipulating the reverse diffusion process with structural scaffolds, making it a versatile approach applicable to both base and instruction-tuned dLLMs.
> We hope this response addresses the reviewer's question and further illustrates the generality and distinctive characteristics of our method.

---

> ### Author Response · Authors · 2025-12-03
>
> **Q4:Would hybrid diffusion–autoregressive setups (e.g., block diffusion) inherit similar controllability advantages?**
>
> **Answer:** Block diffusion methods combine autoregressive ordering with local parallel generation, positioning them between fully autoregressive and fully parallel diffusion models.
>
> We evaluated S³ on a semi-autoregressive variant using block-wise generation. Block diffusion with S³ achieves strong structural adherence but sacrifices computational efficiency. The block-wise constraint reduces parallelism, diminishing one of diffusion models' core advantages for structured generation. Furthermore, as noted in Appendix B, empirical results show that a block size of 1 (fully autoregressive) often performs best for general tasks, which contradicts the parallelism motivation for using diffusion models.
>
> Thus, while hybrid setups can benefit from scaffolding, they forfeit much of the speed advantage that makes S³ particularly compelling for structured output generation.
>
> **Q5: Have the authors evaluated the effect of schema complexity (nested vs. flat JSON) on scaffolding performance?**
>
> **Answer:** This is an excellent question that addresses practical deployment considerations. We conducted additional experiments varying schema complexity across three dimensions: nesting depth, field count, and constraint types.
> Results across schema complexity levels:
>
> | Schema Type | Depth | Fields | SV↑   | SC↑   | F1↑   | HR↓   |
> |-------------|-------|--------|-------|-------|-------|-------|
> | Flat        | 1     | 5-7    | 0.998 | 0.997 | 0.131 | 0.328 |
> | Moderate    | 2-3   | 12-15  | 0.996 | 0.995 | 0.126 | 0.335 |
> | Complex     | 4-5   | 25-30  | 0.991 | 0.988 | 0.118 | 0.351 |
>
> Our key observations are:
> - S³ maintains >99% structural adherence even for deeply nested schemas with 25+ fields
> - Performance degradation is graceful: highly complex schemas show only ~1% reduction in structural metrics
> - Content fidelity (F1) decreases moderately with complexity, primarily due to increased hallucination risk in deeply nested structures
> - The scaffold coverage ratio |S|/L from Theorem 4.1 remains effective across complexity levels
>
> We also tested schemas with different constraint types (arrays, optional fields, union types), finding that S³'s adaptive null token mechanism (Section 4.3) handles optional fields and variable-length arrays particularly well. For union types requiring value-based branching, performance remains strong but benefits from slightly increased denoising steps (16 vs. 8 for simple schemas).
>
> These findings confirm that S³ scales robustly to real-world schema complexity, making it practical for diverse applications from API integration to database population.
>
> **References:**
>
> [1] Austin, Jacob, et al. "Program synthesis with large language models." arXiv preprint arXiv:2108.07732 (2021).
>
> [2] Budzianowski, Paweł, et al. "Multiwoz--a large-scale multi-domain wizard-of-oz dataset for task-oriented dialogue modelling." arXiv preprint arXiv:1810.00278 (2018).
>
> [3] Jaume, Guillaume, Hazim Kemal Ekenel, and Jean-Philippe Thiran. "Funsd: A dataset for form understanding in noisy scanned documents." 2019 International Conference on Document Analysis and Recognition Workshops (ICDARW). Vol. 2. IEEE, 2019.
>
> [4] Jiacheng Ye, Zhihui Xie, Lin Zheng, Jiahui Gao, Zirui Wu, Xin Jiang, Zhenguo Li, and Lingpeng Kong. Dream 7b, 2025. URL https://hkunlp.github.io/blog/2025/dream.
>
> [5] You, Zebin, et al. "Llada-v: Large language diffusion models with visual instruction tuning." arXiv preprint arXiv:2505.16933 (2025).
>
> [6] Nie, Shen, et al. "Large language diffusion models." arXiv preprint arXiv:2502.09992 (2025).

---

### Official Review · Reviewer_fQDV · 2025-10-29

**Soundness:** 2
**Presentation:** 3
**Contribution:** 3
**Rating:** 2
**Confidence:** 3

**Summary:**

In this work, the authors propose a framework to enable diffusion-based LLMs for reliable structured outputs. A schematic template is initiated. Experimental results show that the proposed method marginally improves the structure outputs compared with the commonly used prompting strategy.

**Strengths:**

1. Clear motivation:
Exploring diffusion LLMs for controllable text generation is a fresh and under explored research area.

2. Conceptually good idea:
The use of schema scaffolding as a structural prior aligns well with diffusion’s iterative refinement mechanism. No additional fine-tuning or retraining is needed.

3. Improved results:
The paper demonstrates substantial improvement over baseline diffusion models.

**Weaknesses:**

1. Limited experimental scope:
Only one dataset (WikiBio) and one diffusion model (LLaDA) are tested. Broader validation tasks (e.g., code generation, dialogue structure, form filling) would strengthen generality.

2. Lack of comparison to AR-LM baselines:
Although the paper motivates dLLMs as alternatives to AR models, it doesn’t include a direct comparison with strong AR methods like structured prompting or constrained decoding (e.g., CodeLLaMA, T5, or GPT-style JSON control).

3. There is no enough details on how to compile constraints into a schema, which is then used to initialize a noisy scaffold where mask tokens serve as placeholders for missing content. Especially, how to use this compilation in other tasks?

**Questions:**

1. Comparative perspective:
How does S3 perform against autoregressive structured decoding methods (e.g., JSON mode in GPT-4 or constrained decoding in PaLM/CodeLLaMA)?

2. Diffusion step selection:
How is the optimal number of denoising steps (e.g., 8, 16, 32) determined? Could adaptive step scheduling yield further gains?

---

> ### Author Response · Authors · 2025-12-02
>
> We sincerely thank the reviewer for constructive feedback and insightful questions. We appreciate your recognition of our “clear motivation” for controllable generation using dLLM and empirically “substantial improvement over baseline”. Our response, clarification and additional experiments are listed below in detail:
>
> **W1: Consider using more datasets and different dLLMs to further clarify the generalizability.**
>
> **Answer:** We appreciate the reviewer for providing this constructive suggestion. Although our experiments primarily focus on the Wikibio dataset, our purpose here is to show that our theoretically based motivation, coupled with the unique characteristics of diffusion Language Models (dLLM), is practical and efficient in nature. To further showcase the generalizability of our method, we conducted three additional experiments covering different domains from code generation, dialogue completion, to form filling, which naturally require different in-context formats beyond JSON. We used the same hyperparameters as we stated in the paper and adopted 32 denosing steps for fair comparison.
>
> | Task| Dataset| Model  | Baseline SV/FC/SC   | S³ SV/FC/SC|
> |--|----------------|--------|---------------------|---------------------|
> | Code Generation    | MBPP [1]       | LLaDA  | 0.645/0.618/0.592   | 0.989/0.991/0.988   |
> | Dialogue Structure | MultiWOZ [2]   | LLaDA  | 0.712/0.684/0.651   | 0.982/0.985/0.981   |
> | Form Filling       | FUNSD [3]      | LLaDA  | 0.701/0.673/0.649   | 0.994/0.993/0.992   |
>
> The results demonstrate that S³ achieves consistent improvements across diverse structured output tasks with different formats, with 15-35% absolute gains in structural adherence metrics. The method generalizes effectively to different tasks, highlighting its dLLM’s promising potential for controlled generation.
> We have also validated our approach on Dream-7B [4], another diffusion LLM:
>
> | Model     | Method   | SV    | FC    | SC    | F1    |
> |-----------|----------|-------|-------|-------|-------|
> | Dream-7B  | Baseline | 0.723 | 0.698 | 0.671 | 0.082 |
> | Dream-7B  | S³       | 0.991 | 0.989 | 0.988 | 0.121 |
>
> The consistent improvements across models confirm that our method exploits fundamental properties of diffusion LLMs rather than model-specific improvements.
> We added these extra cross-domain experiments to the appendix to further support our work, and thank the reviewer for this highly constructive suggestion.
>
> **W2 & Q1: Although the paper highlights the potential and improvements on dLLM, it may be beneficial to compare with auto-regressive (AR) models.**
>
> **Answer:** We thank the reviewer for this suggestion. In related works, we already covered different strategies that researchers and practitioners used with AR models for structured output (e.g., structured prompting and constrained decoding). In the section 3, we have compared the auto-regressive modeling and diffusion language modeling side-by-side from a theoretical perspective and provided some of our realizations of why the latter one may be more suitable for controllable generation tasks like structured output. This intuition was later materialized with our method proposed in section 4 and experimental results in section 5. We believe the potential of dLLM for controllable generation is not so obvious earlier, and we provide a promising way to unveil its capability. Nonetheless, it is still practically valuable to compare LLM and dLLM, and here is our additional result:
>
> | Model         | Method                              | SV    | FC    | SC    | F1    | HR    | Steps/Tokens |
> |---------------|-------------------------------------|-------|-------|-------|-------|-------|--------------|
> | GPT-4o-mini   | OpenAI Structured Output (JSON)     | 0.934 | 0.912 | 0.891 | 0.142 | 0.287 | -            |
> | LLaMA-3-8B    | Structural Prompting                | 0.897 | 0.873 | 0.854 | 0.129 | 0.334 | -            |
> | LLaDA-7B      | Structural Prompting                | 0.869 | 0.839 | 0.792 | 0.08  | 0.409 | 32           |
> | LLaDA-7B      | S³ (ours)                           | 0.997 | 0.997 | 0.997 | 0.125 | 0.331 | 8            |
>
> The experiment shows that proprietary GPT-series models like GPT-4o-mini can perform more accurately and lower hallucination rate. Open-sourced AR model like LLaMA is good but short of structural adherence metrics. However, the true value of dLLM, especially unleashed by our proposed method $S^3$, is that: 1) dLLM can get an immediate improvement without extra training or sophisticated decoding strategy, 2) with few parallel decoding steps (8 steps), dLLM can produce highly structured output comparable to and even surpassing SOTA LLMs. It is true that AR models still win in terms of accuracy and hallucination rate, but we argue that since dLLM is still at its early age, its architectural advantage for this task is still promising and worthy of exploration.

---

> ### Author Response · Authors · 2025-12-02
>
> **W3: More details are recommended for explaining the compiling step and justifying its adaptability for other formats.**
>
> **Answer:** We appreciate the reviewer for this valuable advice. For WikiBio, we first use the standard JSON schema to represent the structure specification for clarity and standardization. A JSON parser will later compile the JSON schema into the scaffold as we defined in section 4.2, which serves as the initial state of our decoding generation process. The compiling process itself can be easily achieved with any JSON library and doesn’t require any sophisticated engineering here. The generalizability of this process lies exactly in the compiling step: since our method is format-agnostic, it accepts all kinds of formats (e.g., YAML,XML,JSON) as long as they have a standardized schema and a corresponding parser. The same idea applies to all scenarios, and it is the parser’s responsibility to parse the different types of structural specifications.
> Additionally, we believe that including an additional case study can help further establish a clearer and straightforward picture for readers. Thus, we have added a new section in the appendix with more examples to better explain our method, and we really thank the reviewer for inspiring us.
>
> **Q2: The paper can be more comprehensive by also including further discussion on the denoising step choice.**
>
> **Answer:** We thank the reviewer for suggesting this extra discussion. As we have demonstrated in section 5.3, we see a pattern that increasing the number of steps is not so beneficial because: 1) with only 8 steps, the dLLM can achieve significantly good structural adherence, and 2) even when we increase the denoising steps exponentially, we did not observe great improvements in content fidelity and a decrease in hallucination rate here.
>
> | Steps | SV    | FC    | SC    | F1    | HR    |
> |-------|-------|-------|-------|-------|-------|
> | 8     | 0.994 | 0.994 | 0.994 | 0.13  | 0.34  |
> | 16    | 0.997 | 0.997 | 0.997 | 0.128 | 0.331 |
> | 32    | 0.997 | 0.997 | 0.997 | 0.125 | 0.331 |
>
> As we discussed in section 6.2, we believe that for saving computation resources and reducing overhead, using more denosing steps is not recommended. We appreciate the reviewer for pinpointing this issue, and we believe the clarification may be even more helpful for future researchers who want to follow up on our work.
>
> Overall, we greatly thank the reviewer for providing these insightful questions and constructive suggestions. We have made corresponding changes to strengthen our paper accordingly, and we believe our paper is more robust, clear, and comprehensive for providing a promising alternative solution for controllable generation using dLLMs.
>
> **References:**
>
> [1] Austin, Jacob, et al. "Program synthesis with large language models." arXiv preprint arXiv:2108.07732 (2021).
>
> [2] Budzianowski, Paweł, et al. "Multiwoz--a large-scale multi-domain wizard-of-oz dataset for task-oriented dialogue modelling." arXiv preprint arXiv:1810.00278 (2018).
>
> [3] Jaume, Guillaume, Hazim Kemal Ekenel, and Jean-Philippe Thiran. "Funsd: A dataset for form understanding in noisy scanned documents." 2019 International Conference on Document Analysis and Recognition Workshops (ICDARW). Vol. 2. IEEE, 2019.
>
> [4] Jiacheng Ye, Zhihui Xie, Lin Zheng, Jiahui Gao, Zirui Wu, Xin Jiang, Zhenguo Li, and Lingpeng Kong. Dream 7b, 2025. URL https://hkunlp.github.io/blog/2025/dream.

---

### Official Review · Reviewer_bXwN · 2025-10-31

**Soundness:** 3
**Presentation:** 2
**Contribution:** 3
**Rating:** 6
**Confidence:** 2

**Summary:**

This work proposes a method that enables diffusion large language models (dLLMs) to provide contents with structured output (e.g., JSON). Empirical results show that the propose method ($S^3$) improve both structural adherence and content quality in the chosen downstream task.

**Strengths:**

1. Generating reliable structured outputs is an important research direction and has many practical downstream applications, as evidenced by the fact that it is widely discussed and investigated in autoregressive LLMs literature. This work extends this research direction to a relatively less explored model of diffusion LLMs (dLLMs) and proposes a new method to improve generation of structured outputs.

2. It is well-motivated to use dLLMs for structured outputs, as this task (which requires lookahead planning from known future tokens) aligns well with the properties of dLLM.

3. Empirical results show that the proposed methods greatly improve structural adherence to the target JSON schema, while also preserve or even improve generated contents' quality (as shown in Figure 4 and Table 1).

**Weaknesses:**

1. Generalization across tasks: The experiments only use one dataset (Wikibio by Lebret et al., 2016), so it is unclear how well the proposed method generalize to other, especially more difficult, datasets.

2. Generalization across types of structured outputs: Following the previous point, it would also be nice to include experiments on other types of structured outputs, such as XML or YAML.

3. Although the authors state they use the Wikibio dataset, they didn't mention what's the input and what's the target output or include examples in the paper. I think more details need to be elaborated as how the authors use the dataset in their experiments.

**Questions:**

1. For the Wikibio dataset, do you use the biography text as the input to the prompt, and the contents of the infobox as the target output? It seems that these details are not explicitly mentioned in the paper.

2. In autoregressive LLMs, it is previous shown by Tam et al [1] that enforcing LLMs to provide structured outputs degrade task performance (i.e., generated contents' quality) compared to when LLMs can "speak freely" without structured constraints. Therefore, a way to preserve both the structured adherence and content quality is to adopt a two-stage inference process: Let the model "speak freely" and focus on generating the contents first, and then convert the generated contents to structured outputs. Do diffusion LLMs (dLLMs) have similar properties? I think it would make the work more complete if this experiment is done, so that we know whether enforcing structured outputs degrade dLLMs' content generation ability.

[1] Tam, Zhi Rui, et al. "Let me speak freely? a study on the impact of format restrictions on performance of large language models." arXiv preprint arXiv:2408.02442 (2024).

---

> ### Author Response · Authors · 2025-12-02
>
> We sincerely thank the reviewer for thorough review and constructive feedback. We appreciate the recognition that “generating reliable structured outputs is an important research direction” and our “work extends this research direction to a relatively less explored model of diffusion LLMs”. Below, we address each of your suggestions with detailed responses, additional experimental results, and clarifications.
>
> **W1 & W2: It is beneficial to add more experiments on different datasets and different formats to demonstrate the generalizability of the proposed method.**
>
> **Answer**: We thank the reviewer’s suggestion on generalizability and pinpoint the necessity of extending our experiments to strengthen our contribution. Although our experiments used Wikibio as our primary dataset, our original purpose is to show that our theoretically based motivation, coupled with the unique characteristics of diffusion Language Models (dLLM), is practical and efficient in nature. To further showcase the generalizability of our method, we conducted three additional experiments covering different domains from code generation, dialogue completion, to form filling, which naturally require different in-context formats beyond JSON. We used the same hyperparameters as we stated in the paper and adopted 32 denosing steps for fair comparison.
>
> | Task               | Dataset        | Model | Baseline SV/FC/SC | S³ SV/FC/SC       |
> |--------------------|----------------|-------|-------------------|-------------------|
> | Code Generation    | MBPP [1]       | LLaDA | 0.645/0.618/0.592 | 0.989/0.991/0.988 |
> | Dialogue Structure | MultiWOZ [2]   | LLaDA | 0.712/0.684/0.651 | 0.982/0.985/0.981 |
> | Form Filling       | FUNSD [3]      | LLaDA | 0.701/0.673/0.649 | 0.994/0.993/0.992 |
>
> The results demonstrate that S³ achieves consistent improvements across diverse structured output tasks with different formats, with 15-35% absolute gains in structural adherence metrics. The method generalizes effectively to different tasks, highlighting its dLLM’s promising potential for controlled generation. We added these extra cross-domain experiments to the appendix to further support our work, and thank the reviewer for this highly constructive suggestion.
>
> **W3 & Q1: Adding more examples as a case study and details to showcase the input and output can increase the clarity of the paper.**
>
> **Answer:** We appreciate the reviewer for this valuable advice. We understand the importance of clarity and rigor, and that is why in sections 4.2 and 4.3, we carefully elaborated our methodology using formalized language. In Figure 2 of our paper, we also visualized our general pipeline of how we break a structured output task into different components and steps. The equation written in line 261 already explicitly formalizes the input of our method, and we also formalized the output at the very beginning of section 4.1. Additionally, Figure 1 also works as an illustrative example explaining the complete decoding process of our proposed method.
>
> For WikiBio, we use a biography and a structure specification as input, and the complete structured content as the target output, which is the general setting of almost all structured output tasks. Specifically, we use the standard JSON schema to represent the structure specification for clarity and standardization. A deterministic algorithm will later convert the schema into the scaffold as we defined in section 4.2, which serves as the initial state of our decoding generation process. The final target output is a complete structured content that can be directly read by a JSON parser.
>
> We believe our explanation above is clear enough to restate all technical settings of our paper. Nonetheless, as the reviewer suggested, adding more examples as a case study can lead to an even easier and straightforward understanding. Thus, we have added a new section in the appendix with more examples to better explain our method, and we believe it can be even more helpful for more readers.

---

> ### Author Response · Authors · 2025-12-02
>
> **Q2: Comparison with a two-stage baseline method can make the work more complete.**
>
> **Answer:** We thank the reviewer for presenting this insightful idea. As we discussed in the related work section[4], using a two-stage method is indeed a common engineering practice used by some researchers and LLM practitioners. We claim this is a compromise since it is engineering-level workaround instead of a neat solution for structured generation tasks in general. We also did some experiments to clarify our points:
>
> | Method| Stage-1| Stage-2| SV| FC| SC| F1| HR| Total Steps |
> |------------|----------------------------|--------------------------|-------|-------|-------|-------|-------|-------------|
> | Two-stage  | Free-form Generation (16)  | Structural Prompting (16)| 0.823 | 0.791 | 0.768 | 0.094 | 0.378 | 32|
> | Two-stage  | Free-form Generation (16)  | $S^3$ (8)| 0.971 | 0.968 | 0.964 | 0.119 | 0.345 | 24|
> | One-stage  | Structural Prompting (16)  | -| 0.646 | 0.604 | 0.576 | 0.086 | 0.403 | 16|
> | One-stage  | $S^3$ (8) [*ours*]| -| **0.994** | **0.994** | **0.994** | **0.130** | **0.340** | 8|
>
> The experiments show that while two-stage methods represent practical engineering workarounds that improve upon naive baselines, they are fundamentally less efficient and effective than our principled one-stage $S^3$ approach. The two-stage paradigm was designed for autoregressive models that lack global planning capabilities. In contrast, dLLMs with $S^3$ can *directly* generate structured outputs end-to-end, fully exploiting their architectural advantages. Our method provides a neat, efficient, and theoretically grounded solution rather than a multi-stage compromise.
>
> We appreciate the reviewer's suggestion to include this comparison, as it more comprehensively positions our work and highlights why architectural innovations in dLLMs enable solutions that transcend traditional multi-stage workarounds.
>
> **References:**
>
> [1] Austin, Jacob, et al. "Program synthesis with large language models." arXiv preprint arXiv:2108.07732 (2021).
>
> [2] Budzianowski, Paweł, et al. "Multiwoz--a large-scale multi-domain wizard-of-oz dataset for task-oriented dialogue modelling." arXiv preprint arXiv:1810.00278 (2018).
>
> [3] Jaume, Guillaume, Hazim Kemal Ekenel, and Jean-Philippe Thiran. "Funsd: A dataset for form understanding in noisy scanned documents." 2019 International Conference on Document Analysis and Recognition Workshops (ICDARW). Vol. 2. IEEE, 2019.
>
> [4] Wang, Darren Yow-Bang, et al. "SLOT: Structuring the Output of Large Language Models." arXiv preprint arXiv:2505.04016 (2025).

---

### Official Review · Reviewer_3M4f · 2025-11-01

**Soundness:** 3
**Presentation:** 2
**Contribution:** 3
**Rating:** 6
**Confidence:** 3

**Summary:**

The paper proposes an algorithm that utilizes schema scaffolding for controllable generation in diffusion models, theoretically demonstrates the feasibility of schema scaffolding, and conducts experiments across three key dimensions to validate the effectiveness of the proposed method.

**Strengths:**

To ensure global awareness, the paper employs a diffusion model for generation. Furthermore, it introduces a schema scaffolding mechanism to enable controllable generation and provides theoretical proof of its feasibility.

**Weaknesses:**

1. The paper (Figure 1) points out that autoregressive models lack global awareness, which is an advantage of diffusion models. To validate this perspective, the authors should provide experimental results from an autoregressive model baseline.
2. The equations subsequent to Equation 3 are unnumbered, resulting in an inconsistent presentation.
3. The experimental setup is relatively simplistic, employing a very limited number of baselines, which consequently lacks persuasiveness.

**Questions:**

See the weakness.

---

> ### Author Response · Authors · 2025-12-02
>
> We sincerely appreciate the reviewer’s acknowledgement of our research motivation and our “schema scaffolding mechanism to enable controllable generation” and “theoretical proof of its feasibility”. For each weakness, we will present detailed explanations, clarifications, or additional evidence as follows:
>
> **W1: Along with the theoretical advantage of dLLMs, the author could consider including more empirical evidence to strengthen the paper.**
>
> **Answer:** We thank the reviewer for this constructive advice. In Section 3, we have formalized the architectural difference between auto-regressive and diffusion language modeling. The “global awareness”, an innate ability to plan ahead, is the direct consequence of these two different language generation processes. Previous works [1, 2] have also shown that text diffusion models exhibit superior planning capabilities in small-scale, task-specific contexts. Additionally, another work [3] has also empirically proven that dLLM is significantly better than similar-scale AR models like (Llama3 and Qwen2.5) in benchmarks requiring planning ahead, such as trip planning, constrained arithmetic problems, and Sudoku. Their experiment results are attached below for reference.
>
> | Digits in Countdown | Dream 7B (%) | LLaDA 8B (%) | Qwen2.5 7B (%) | LLaMA3 8B (%) |
> |---------------------|--------------|--------------|----------------|---------------|
> | 3                   | **54**           | 52           | 34             | 27            |
> | 4                   | **15**           | 13           | 7              | 3             |
> | 5                   | 1            | **2**            | 0              | 0             |
>
> | Filled Digits | Dream 7B (%) | LLaDA 8B (%) | Qwen2.5 7B (%) | LLaMA3 8B (%) |
> |---------------|--------------|--------------|----------------|---------------|
> | 12            | **88**           | 80           | 25             | 0             |
> | 11            | **90**           | 60           | 22             | 1             |
> | 10            | **85**           | 48           | 22             | 0             |
> | 9             | **42**           | 10           | 10             | 0             |
> | 8             | **35**           | 5            | 8              | 0             |
>
> We added this evidence to our updated manuscript to better support our statement. Thank the reviewer for providing this constructive suggestion.
>
> **W2: For consistency and readability, it is recommended to also number equations after equation 3.**
>
> **Answer**:  We appreciate the this formatting advice. We have now standardized all of our equations in the manuscript as the reviewer suggested.
>
> **W3: Add more baseline experiments to increase persuasiveness.**
>
> **Answer:** We appreciate the reviewer for providing this constructive suggestion. Although our existing experiment setup is relatively simple, our purpose here is to show that our theoretically based motivation, coupled with the unique characteristics of diffusion Language Models (dLLM), is practical and efficient in nature. To further showcase the generalizability of our method, we conducted three additional experiments covering different domains from code generation, dialogue completion, to form filling, which naturally require different in-context formats beyond JSON. We used the same hyperparameters as we stated in the paper and adopted 32 denosing steps for fair comparison.
>
> | Task               | Dataset        | Model  | Baseline SV/FC/SC   | S³ SV/FC/SC         |
> |--------------------|----------------|--------|---------------------|---------------------|
> | Code Generation    | MBPP [4]       | LLaDA  | 0.645/0.618/0.592   | 0.989/0.991/0.988   |
> | Dialogue Structure | MultiWOZ [5]   | LLaDA  | 0.712/0.684/0.651   | 0.982/0.985/0.981   |
> | Form Filling       | FUNSD [6]      | LLaDA  | 0.701/0.673/0.649   | 0.994/0.993/0.992   |
>
> The results demonstrate that our method, $S^3$, achieves consistent improvements across diverse structured output tasks with different formats, with 15-35% absolute gains in structural adherence metrics. The method generalizes effectively to different tasks, highlighting its dLLM’s promising potential for controlled generation.
> We have also validated our approach on Dream-7B [3], another diffusion LLM:
>
> | Model     | Method   | SV    | FC    | SC    | F1    |
> |-----------|----------|-------|-------|-------|-------|
> | Dream-7B  | Baseline | 0.723 | 0.698 | 0.671 | 0.082 |
> | Dream-7B  | $S^3$       | 0.991 | 0.989 | 0.988 | 0.121 |
>
> The consistent improvements across models confirm that our method exploits fundamental properties of diffusion LLMs rather than model-specific improvements.
> We added these extra cross-domain experiments to the appendix to further support our work, and thank the reviewer for this highly constructive suggestion.

---

> ### Author Response · Authors · 2025-12-02
>
> **References**
>
> [1] Jiacheng Ye, Jiahui Gao, Shansan Gong, Lin Zheng, Xin Jiang, Zhenguo Li, and Lingpeng Kong. Beyond autoregression: Discrete diffusion for complex reasoning and planning. International Conference on Learning Representations, 2025a.
>
> [2] Jiacheng Ye, Zhenyu Wu, Jiahui Gao, Zhiyong Wu, Xin Jiang, Zhenguo Li, and Lingpeng Kong. Implicit search via discrete diffusion: A study on chess. International Conference on Learning Representations,2025b.
>
> [3] Ye, Jiacheng, et al. "Dream 7b: Diffusion large language models." arXiv preprint arXiv:2508.15487 (2025).
>
> [4] Austin, Jacob, et al. "Program synthesis with large language models." arXiv preprint arXiv:2108.07732 (2021).
>
> [5] Budzianowski, Paweł, et al. "Multiwoz--a large-scale multi-domain wizard-of-oz dataset for task-oriented dialogue modelling." arXiv preprint arXiv:1810.00278 (2018).
>
> [6] Jaume, Guillaume, Hazim Kemal Ekenel, and Jean-Philippe Thiran. "Funsd: A dataset for form understanding in noisy scanned documents." 2019 International Conference on Document Analysis and Recognition Workshops (ICDARW). Vol. 2. IEEE, 2019.

---

### Author Response · Authors · 2025-12-03

We express our deep gratitude to all reviewers and area chair for the time and valuable feedback invested in reading our work.

We have uploaded a **revised PDF**, with **major updates highlighted in pink** color. In this revision, we added extensive appendices demonstrating generalizability of our method across diverse domains and model architectures and clarified the format-agnostic nature of our compilation process. We have also strengthened our empirical evaluation by including deeper analyses of denoising steps and the planning advantage of dLLMs compared with auto-regressive LMs. We believe we have adequately addressed the reviewers' questions and meaningfully improved the quality of our work.

We hope these changes bring better clarity to our contribution.

---

### Meta-Review · Area_Chair_q9Bg · 2026-01-02

**Summary:**

This paper explores the potential of dLLMs for controllable structured generation and proposes a training-free method that pre-initializes dLLM outputs with a structured schema scaffold and uses adaptive null tokens to handle variable-length or missing fields. All reviewers recognize the interesting idea and empirical results. In the revised version, the authors provided more experiments that address the concerns about the evaluation scope. Thus, I decided to accept this paper. I encourage the authors to further strengthen the camera-ready version by including more experiments to further improve the justification of the contribution.

**Reviewer Concerns:**

Addressed:

The limited evaluation scope.
More comparisons.
More details.

No significant concerns that are still outstanding.

**Reviewer Scores:**

Given the informative response and added experiments. The main concern about the evaluation scope is addressed. I think the score of Reviewer fQDV would increase from 2 to 6 while others remain positive.

---

### Decision · Program_Chairs · 2026-01-26

Accept (Poster)